# Novel Paintings from the Latent Diffusion Model through Transfer Learning

Dayin Wang [1], Chong Ma [2] and Siwen Sun [3,*]

1 Space Star Technology Co., Ltd., Beijing 100086, China; wdymc@163.com
2 Space Innovation Technology Co., Ltd., Beijing 100070, China; mbyd3236@163.com
3 Institute of Intelligent Manufacturing, Heilongjiang Academy of Sciences, Harbin 150090, China
* Correspondence: siwensun1987@126.com

**Abstract:** With the development of deep learning, image synthesis has achieved unprecedented achievements in the past few years. Image synthesis models, represented by diffusion models, demonstrated stable and high-fidelity image generation. However, the traditional diffusion model computes in pixel space, which is memory-heavy and computing-heavy. Therefore, to ease the expensive computing and improve the accessibility of diffusion models, we train the diffusion model in latent space. In this paper, we are devoted to creating novel paintings from existing paintings based on powerful diffusion models. Because the cross-attention layer is adopted in the latent diffusion model, we can create novel paintings with conditional text prompts. However, direct training of the diffusion model on the limited dataset is non-trivial. Therefore, inspired by the transfer learning, we train the diffusion model with the pre-trained weights, which eases the training process and enhances the image synthesis results. Additionally, we introduce the GPT-2 model to expand text prompts for detailed image generation. To validate the performance of our model, we train the model on paintings of the specific artist from the dataset *WikiArt*. To make up for missing image context descriptions of the *WikiArt* dataset, we adopt a pre-trained language model to generate corresponding context descriptions automatically and clean wrong descriptions manually, and we will make it available to the public. Experimental results demonstrate the capacity and effectiveness of the model.

**Keywords:** image synthesis; latent space; transfer learning; text prompt expansion

## 1. Introduction

Image synthesis has made excellent achievements with the rapid progress of deep learning and computer vision, which has a wide application in designing and painting. Generative adversarial network [1] made great progress in generating high-quality images. It is inspired by the game theory: the generator and the discriminator are competing with each other, which makes both evolve at the same time. However, due to the adversarial training nature, the training of the adversarial network is known as unstable and the diversity of the generated image is limited. The generative model, variational autoencoder [2], is similar to the autoencoder but deeply rooted in the variational Bayesian and graphical model. The autoencoder is the neural network, which is designed to learn the identity function to reconstruct the input in an unsupervised way. Different from the autoencoder, the variational autoencoder maps the original input into a distribution, whose training process is supervised by the Kullback–Leibler divergence loss function.

Recently, diffusion probabilistic models [3–7], are built from the denoising autoencoder, which has shown unprecedented results in image synthesis, super-resolution, inpainting, and stylization. Compared with generative adversarial models, the diffusion model with likelihood nature exhibits more stable training and models complicated structures of the image by exploiting shared parameters. There are two Markov chains, the forward diffusion process and the reverse denoising process, in the diffusion model. The forward diffusion

process adds Gaussian random noise to the given image in sequence until the disturbed sample satisfies the Gaussian distribution. The reverse denoising process generates the image from the Gaussian noise conditioned by the given input, e.g., text, audio, and image. The forward diffusion destroys the image with random noises, and the reverse denoising learns to reconstruct the image gradually. The diffusion model with mode-covering characteristics has a powerful ability to represent imperceptible details. However, repeated functions and gradient calculations in the diffusion model demand massive computing resources. Training the diffusion model [7] requires plenty of computational resources and takes several hundred GPU days, which is quite costly and difficult for common use. Inference and evaluation of the trained model are also expensive in memory and time.

To ease massive computing resource consumption and improve the accessibility of the diffusion model, Rombach et al. [6] proposes the latent diffusion model. The latent diffusion model reduces the computational complexity without obvious performance degradation. The latent diffusion model accomplishes this by learning the denoising process in latent space, striking a balance between visual quality and computational complexity. Furthermore, the diffusion model incorporates a cross-attention framework, enabling free image synthesis, which can accommodate various input modalities, e.g., text, audio, and images. Finally, the decoder of the autoencoder is used to translate the denoised latent code into the real image. Building upon the powerful latent diffusion model, we are dedicated to creating innovative paintings. Throughout history, numerous talented artists, such as Leonardo da Vinci and Vincent van Gogh, have created many impressive artworks in their careers. Our aspiration is to invite those late talented artists to create novel paintings on modern topics with the diffusion model. We attempt to train the latent diffusion model on paintings of the specific artist from the dataset *WikiArt* [8]. However, these late artists' collections offer fewer examples compared to the extensive training data required for neural networks. Therefore, inspired by transfer learning, we train the latent diffusion model with publicly available pre-trained weights. The pre-trained diffusion model has already processed millions of images and demonstrated excellent representation capabilities, which greatly benefit our model's training process. After retraining, we can synthesize the novel painting with any given text prompt. To the best of our knowledge, we propose to create novel paintings from famous artists' works based on the diffusion model for the first time.

For detailed image generation, we incorporate the large language model, GPT-2 [9], to enhance the text prompts. The prompt is put into the pre-trained large language model with rich knowledge to understand, explain, and extend the prompt further, which supports generating more exquisite images. Because the existing dataset *WikiArt* lacks the corresponding image context description, we adopt the image tagging model and vision-language model to generate these descriptions, and then we use the contrastive model to remove incorrect image text descriptions. We will release the image context description dataset to the public. Experimental results show the high quality of generated paintings and the effectiveness of our framework. The main contributions of our paper are summarized as follows:

- We propose the painting model to create novel paintings from those late famous artists' works for the first time, which is based on the latent diffusion model with transfer learning.
- We propose the text prompt expansion, which utilizes positives of large language models for completing the text prompts and generating more detailed images.
- We contribute missing image context descriptions, which are complementary to the original *WikiArt* dataset, and we will release it to the public.
- We demonstrate photo-realistic painting results by giving different text prompt inputs to the trained model.

## 2. Related Work

### 2.1. Generative Adversarial Network

Generative adversarial network [1] has demonstrated excellent results in many tasks, such as images, text, and audio. Inspired by the game theory, the generative adversarial network contains the generative model and the discriminative model. The generative model is responsible for modeling the distribution of input data, and the discriminative model is in charge of distinguishing whether the sample data come from ground truth or the generative model. The generative model and the discriminative model are trained and compete simultaneously. The ideal conditions for training are that the generative model captures the data distribution totally, and the discriminative model cannot distinguish whether the sample data come from ground truth or the generative model.

In the image synthesis task, the generative model synthesizes the image from the noise, which is trained to recover image distributions and generate real images. The schematic graph of the generative adversarial network is shown in Figure 1. The discriminative model estimates the probability of the image from ground truth, which works as a critic and tells the fake image from the real image. The generative model tries to trick the discriminative model and synthesize the real image as much possible. The discriminative model is trying to distinguish whether the image is real or fake. The competing game between the generative model and the discriminative model improves their performance simultaneously. Although some generative methods [10–12] achieve impressive image synthesis results, the training of the generative adversarial network is unstable and hard to converge. During training, the generative model may collapse and become stuck in the space where it produces similar failed results. Therefore, many researchers devote themselves to improving the stability of adversarial training.

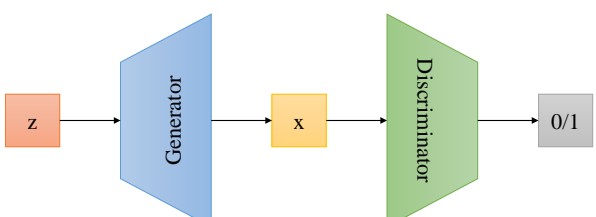

**Figure 1.** The generative adversarial network.

Salimans et al. [13] proposed multiple architectural features and training procedures, feature matching, minibatch discrimination, and virtual batch normalization to stabilize the training of the adversarial network, which achieved high visual quality. Radford et al. [14] experimentally enumerated architectures of the generative and discriminative model, and finally found the best network architecture setting. However, it is a temporary solution and does not solve the problem completely. Arjovsky et al. [15,16] presented a novel generative model, named Wasserstein gan, which replaced Kullback–Leibler divergence with Wasserstein distance, and disposed of unstable training and model collapse to an extent.

### 2.2. Variational Autoencoder

Unlike the autoencoder that maps the input to the fixed vector, variational autoencoder [2], as the generative model, is the variational Bayesian probabilistic model, which translates the input into the distribution and decodes the distribution into the original input. The schematic graph of the variational autoencoder is shown in Figure 2. In the training process, the Kullback–Leibler divergence loss is used to penalize the distance between the estimated posterior and the real posterior. Higgins et al. [17] presented a modified variational autoencoder, named beta-variational autoencoder, which learned disentangled latent representations from original input in an unsupervised manner, which is more stable compared with the generative adversarial network.

Van et al. [18] proposed a vector quantized-variational autoencoder, which was a simple but powerful generative model and was used to learn prior over discrete latent space. Similar to the k-nearest neighbors algorithm, vector quantization is used to map the k-dimension vector into the finite set of code vectors. Razavi et al. [19] proposed the two-level hierarchical vector quantized-variational autoencoder with the self-attention autoregressive model which consists of two stages, training the hierarchical vector quantized-variational autoencoder and learning a prior over the latent discrete codebook. Because the two-level hierarchical vector quantized-variational autoencoder relies on discrete latent variables in a hierarchical setting, the quality of the generated image is excellent.

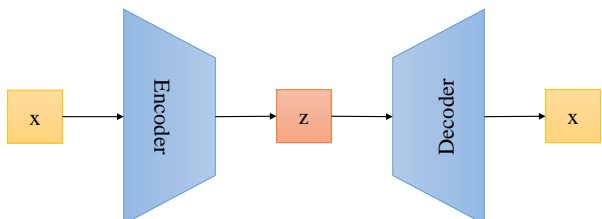

**Figure 2.** The variational autoencoder.

Ramesh et al. [20] learned joint distribution between pair image and text with transformer by extending the vector quantized-variational autoencoder, which achieves impressive results in the zero-shot task. Rombach et al. [21] proposed the conditionally invertible network to relate different representations, provide generic transfer diverse domains, and enable free content synthesis through modification in other domains. Gregor et al. [22] introduced the temporal difference-variation autoencoder, which learned representations over sequential data with temporal difference learning of reinforcement learning.

*2.3. Flow-Based Generative Models*

Neither the generative adversarial model nor the variational autoencoder learns the probability density distribution function of inputs, which is non-trivial to learn. To conquer this problem, Rezende et al. [23] introduced a strong statistics model, normalizing flow, for density prediction, which makes many tasks possible, such as sampling new data points, density estimation, inferring latent variables, and filling incomplete data. In contrast to generative adversarial models and variational autoencoder, the flow-based generative model consists of a sequence of inverse transformation functions and learns the data distribution explicitly, which is demonstrated in Figure 3. The Gaussian distribution is usually used in many latent generative models. However, the distribution of realistic data is much more complicated than the Gaussian distribution. Therefore, the normalizing flow is proposed for better distribution estimation. The normalizing flow transforms the simple distribution into a complex distribution through a sequence of invertible transformations. The flow-based generative model adopts the negative log-likelihood function as the loss function, which makes the input data tractable. Su et al. [24] proposed the f-variational autoencoder integrating normalizing flows and variational autoencoder. F-variational autoencoder improved variational autoencoder with conditional flows, which alleviated image blur and enabled fast inference.

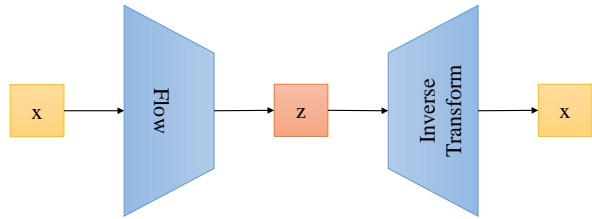

**Figure 3.** The flow-based generative models.

### 2.4. Diffusion Models

Diffusion models [3–5,7] have emerged as powerful tools that synthesize photo-realistic results. Ho et al. [4] proposed denoising diffusion probabilistic models consisting of a sequence of latent variable models, which achieved high-quality image synthesis. However, they need many steps to simulate a Markov chain, which is time-consuming. To accelerate sampling, Song et al. [5] presented denoising diffusion implicit models, which constructed a sequence of efficient non-Markovian implicit denoising processes for the trade-off between sample quality and running speed. To ease the computing resources and improve running speed further, Rombach et al. [6] proposed to learn the denoising process in latent space instead of pixel space. Motivated by the observation that most bits of the image contribute to non-perceptual details, they decomposed the diffusion model into semantic compression and perceptual compression for lightweight computations.

The four types of generative models, generative adversarial networks, variational autoencoders, flow-based models, and diffusion models, have demonstrated great performance in image synthesis. But all of them have their own negatives. The generative adversarial network is well-known for its instability. Variational autoencoder lacks image diversity. Flow-based models rely greatly on specialized architectures for reversible transformations. The Diffusion model achieves unprecedented image synthesis results through a sequence of denoising steps, but it is resource-consuming and time-consuming in training and inference. The latent diffusion model learns the diffusion process in latent space, which eases the required computing resources. Based on above discussions, we adopt the latent diffusion model as the base image synthesis model due to its great performance.

## 3. Materials and Methods

### 3.1. Denoising Diffusion Probabilistic Model

Ho et al. [4] proposed the denoising diffusion probabilistic model. There are two processes, the diffusion process and denoising process, in this model, which is shown in Figure 4. In the diffusion process, given the sample data $x_0$ from the data distribution $q(x)$, the Gaussian noise is appended to the sample data $x_0$ to acquire a sequence of disturbed samples $x_0, x_1, \cdots, x_T$:

$$q(x_t \mid x_{t-1}) = \mathcal{N}\left(x_t; \sqrt{1 - \beta_t} x_{t-1}, \beta_t \mathbf{I}\right),$$
$$q(x_{1:T} \mid x_0) = \prod_{t=1}^{T} q(x_t \mid x_{t-1}), \tag{1}$$

where $T$ means the diffusion steps, $\beta_t$ is the hyper-parameter with range from 0 to 1, $\mathcal{N}(\mu, \Sigma)$ is the Gaussian distribution function with the mean $\mu$ and variance $\Sigma$, $\mathbf{I}$ is the identity matrix. We can obtain the arbitrary sample $x_t$ in the diffusion process based on Equation (1) through the reparameterization trick:

$$
\begin{aligned}
x_t &= \sqrt{\alpha_t} x_{t-1} + \sqrt{1 - \alpha_t} \epsilon_1 \\
&= \sqrt{\alpha_t \alpha_{t-1}} x_{t-2} + \sqrt{1 - \alpha_t} \epsilon_1 + \sqrt{\alpha_t(1 - \alpha_{t-1})} \epsilon_2 \\
&= \sqrt{\alpha_t \alpha_{t-1}} x_{t-2} + \sqrt{1 - \alpha_t \alpha_{t-1}} \bar{\epsilon}_2 \\
&= \dots \\
&= \sqrt{\bar{\alpha}_t} x_0 + \sqrt{1 - \bar{\alpha}_t} \epsilon_t,
\end{aligned}
\tag{2}
$$

where $\alpha_t = 1 - \beta_t$, $\bar{\alpha}_t = \prod_{i=1}^{t} \alpha_i$, and $\epsilon_1, \epsilon_2, \bar{\epsilon}_2, \epsilon_t \sim \mathcal{N}(0, \mathbf{I})$. Usually, $\beta_1 < \beta_2 < \cdots < \beta_T$ and $\bar{\alpha}_1 > \cdots > \bar{\alpha}_T$. $x_T$ will slide into the isotropic Gaussian distribution when $T \rightarrow \infty$. Equation (2) can also be formulated as

$$q(x_t \mid x_0) = \mathcal{N}\left(x_t; \sqrt{\bar{\alpha}_t} x_0, (1 - \bar{\alpha}_t)\mathbf{I}\right). \tag{3}$$

In the denoising process, to reconstruct the image from Gaussian noise $x_T$, we need to reverse the diffusion process and learn the model $p_\theta$ to estimate these conditional probabilities:

$$p_\theta(x_{t-1} \mid x_t) = \mathcal{N}(x_{t-1}; \boldsymbol{\mu}_\theta(x_t, t), \boldsymbol{\Sigma}_\theta(x_t, t)),$$
$$p_\theta(x_{0:T}) = p(x_T) \prod_{t=1}^{T} p_\theta(x_{t-1} \mid x_t). \tag{4}$$

The reverse conditional probability $q(x_{t-1} \mid x_t, x_0)$ can be regarded as the reference to learn $p_\theta(x_{t-1} \mid x_t)$, which is tractable and can be computed by Bayes' rule:

$$
\begin{aligned}
q(x_{t-1} \mid x_t, x_0) &= \mathcal{N}\left(x_{t-1}; \tilde{\boldsymbol{\mu}}_t(x_t, x_0), \tilde{\beta}_t \mathbf{I}\right) \\
&= q(x_t \mid x_{t-1}, x_0) \frac{q(x_{t-1} \mid x_0)}{q(x_t \mid x_0)} \\
&\propto \exp\left(-\frac{1}{2}\left(\frac{(x_t - \sqrt{\alpha_t} x_{t-1})^2}{\beta_t} + \frac{(x_{t-1} - \sqrt{\bar{\alpha}_{t-1}} x_0)^2}{1 - \bar{\alpha}_{t-1}} - \frac{(x_t - \sqrt{\bar{\alpha}_t} x_0)^2}{1 - \bar{\alpha}_t}\right)\right) \\
&= \exp\left(-\frac{1}{2}\left(\left(\frac{\alpha_t}{\beta_t} + \frac{1}{1 - \bar{\alpha}_{t-1}}\right) x_{t-1}^2 - \left(\frac{2\sqrt{\alpha_t}}{\beta_t} x_t + \frac{2\sqrt{\bar{\alpha}_{t-1}}}{1 - \bar{\alpha}_{t-1}} x_0\right) x_{t-1} + C\right)\right) \\
&= \exp\left(-\frac{1}{2\tilde{\beta}_t}(x_{t-1} - \tilde{\boldsymbol{\mu}}_t)^2\right),
\end{aligned}
\tag{5}
$$

where $C$ is the function not relating $x_{t-1}$, $\tilde{\boldsymbol{\mu}}_t$ relies on the $x_t$ and $\epsilon_t$, $\tilde{\beta}_t$ is the scalar:

$$
\begin{aligned}
\tilde{\boldsymbol{\mu}}_t &= \frac{1}{\sqrt{\alpha_t}}\left(x_t - \frac{1 - \alpha_t}{\sqrt{1 - \bar{\alpha}_t}} \epsilon_t\right), \\
\tilde{\beta}_t &= \frac{1 - \bar{\alpha}_{t-1}}{1 - \bar{\alpha}_t} \beta_t.
\end{aligned}
\tag{6}
$$

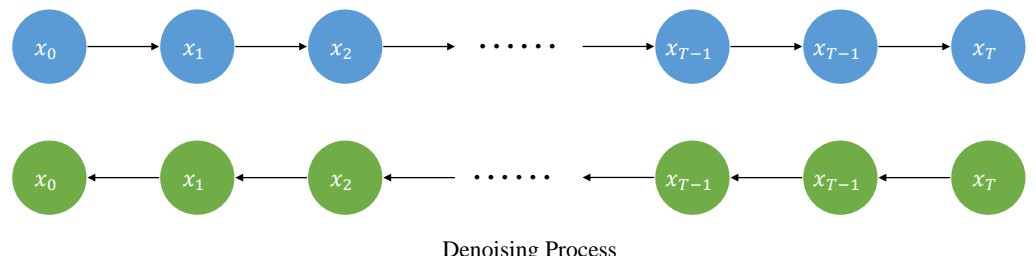

**Figure 4.** The diffusion and denoising process.

The diffusion model is the likelihood model, which learns to denoise the variable with a sequence of denoising autoencoders. To supervise the training of the diffusion model, Ho et al. [4] found that the simplified objective function achieves a better performance:

$$
\begin{aligned}
\mathcal{L}_t^{\mathrm{DM}} &= \mathbb{E}_{x_0, t \sim [1,T], \epsilon_t \sim \mathcal{N}(0, \mathbf{I})}\left[\|\epsilon_t - \epsilon_\theta(x_t, t)\|^2\right] \\
&= \mathbb{E}_{x_0, t \sim [1,T], \epsilon_t \sim \mathcal{N}(0, \mathbf{I})}\left[\left\|\epsilon_t - \epsilon_\theta\left(\sqrt{\bar{\alpha}_t} x_0 + \sqrt{1 - \bar{\alpha}_t} \epsilon_t, t\right)\right\|^2\right],
\end{aligned}
\tag{7}
$$

where the conditional denoising nethwork $\epsilon_\theta$ with parameter $\theta$ is designed to predict the noise from disturbed $x_t$.

### 3.2. Latent Diffusion Model

To ease the demanding computing resources, we train the diffusion model in latent space instead of pixel space. In the latent diffusion model, the learning of the likelihood

model is divided into perceptual compression and semantic compression. The perceptual compression learns the partial semantic variation and removes the high-frequency details. The semantic compression learns the contextual and semantic features of the image. Because most bits of the digital image contribute to eye-imperceptible details, the latent diffusion model is dedicated to searching a computationally cheap but perceptually equivalent space for high-fidelity image synthesis. The structure of the latent diffusion model is shown in Figure 5. The perceptual compression depends on an autoencoder model $\mathcal{E}$, which maps the image $x_0 \in \mathbb{R}^{H \times W \times 3}$ into the latent feature $\mathbf{z}_0 = \mathcal{E}(x_0) \in \mathbb{R}^{h \times w \times c}$ with the downsampling rate $f = H/h = W/w = 2^m, m \in \mathbb{N}$.

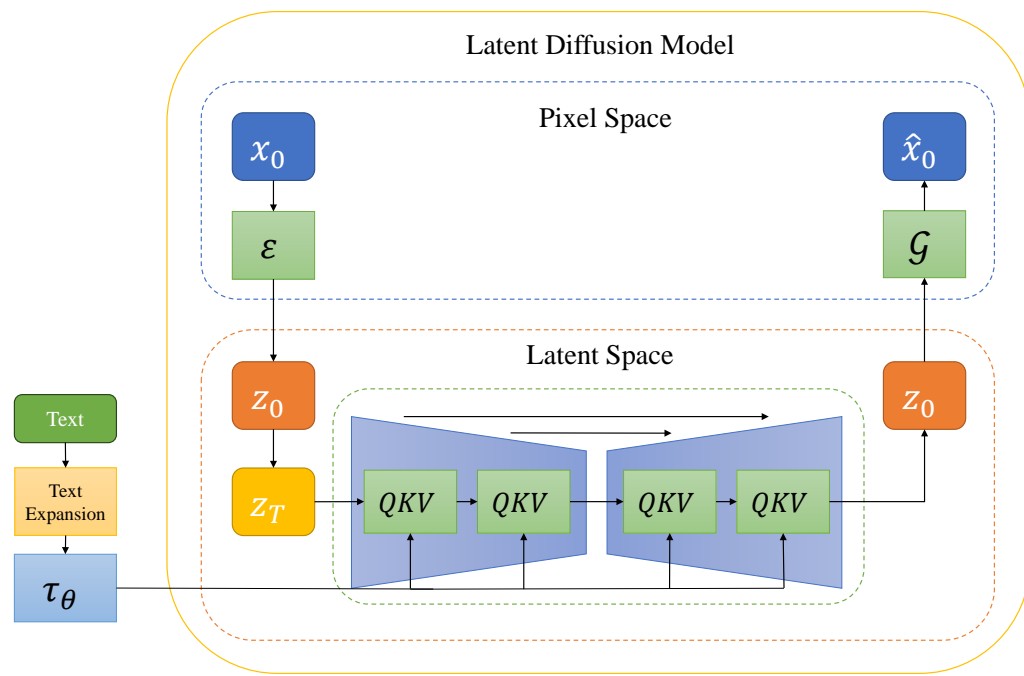

**Figure 5.** The overview of framework.

### 3.2.1. Autoencoder with Regulation

The autoencoder is pre-trained to encode the image $x_0$ into latent feature $\mathbf{z}_0$ and decode the latent feature back to the original image. The self-attention block and residual neural network are adopted in both the encoder $\mathcal{E}$ and the decoder $\mathcal{G}$, to combine the expressiveness of the transformer and the effectiveness of the convolutional neural network. To avoid the arbitrarily high variance of the latent feature, the vector quantization layer and Kullback–Leibler Divergence regulation are adopted in the autoencoder to penalize the latent feature. The loss of the autoencoder is as follows:

(1) The reconstruction loss:

$$\hat{x}_0 = \mathcal{G}(\mathbf{q}(\mathcal{E}(x_0)))$$
$$\mathcal{L}_d = \|x_0 - \hat{x}_0\|^2, \tag{8}$$

where $\mathbf{q}(\cdot)$ is the vector quantization layer.

(2) The Kullback–Leibler Divergence loss on the latent feature $\mathbf{q}(\mathcal{E}(x))$:

$$\mathcal{L}_{KL} = \sum_{c,h,w} \frac{(\mu^2 + \sigma^2 - 1 - \log \sigma^2)}{2}, \tag{9}$$

where $\mu$, $\sigma^2$ are the mean and variance of the latent feature, respectively.

(3) The adversarial loss:

$$\mathcal{L}_G = -\mathbb{E}[D(\hat{x}_0)], \tag{10}$$

$$\mathcal{L}_D = \mathbb{E}[ReLU(1 - D(x_0))] + \mathbb{E}[ReLU(1 + D(\hat{x}_0))], \tag{11}$$

where $D$ is the discriminator, which is employed to penalize the distribution difference between the ground truth and predicted image and enhance the visual quality of the image.

3.2.2. Conditional Denoising Nethwork

Similar to existing generative models, e.g., GAN and VAE, the diffusion model with the conditional encoder is able to model conditional distribution. With the conditional encoder encoding text, audio, semantic map, and pose [25], we gain control over the image synthesis process easily. To integrate the conditional encoder with the diffusion model and fuse different modality inputs effectively, the cross-attention mechanism [26] with U-Net structure [6] makes up for the conditional denoising network, which predicts the noise $\epsilon_t$. To process different modalities, the domain-specific encoder $\tau_\theta$ is adopted, which maps signal $y$ to the latent feature $\tau_\theta(y)$. The cross-attention block performs fusion on $\varphi_i(z_t)$ and $\tau_\theta(y)$, which is able to handle flexible conditional image synthesis. The cross-attention layer is implemented by

$$
\begin{aligned}
Q &= W_Q^{(i)} \cdot \varphi_i(z_t), \\
K &= W_K^{(i)} \cdot \tau_\theta(y), \\
V &= W_V^{(i)} \cdot \tau_\theta(y), \\
\text{Attention} &= \text{softmax}\left(\frac{QK^T}{\sqrt{d}}\right) \cdot V,
\end{aligned}
\tag{12}
$$

where $\varphi_i(z_t)$ means the intermediate feature of the U-Net structure, and $W_Q^{(i)}, W_K^{(i)}, W_V^{(i)}$ are learnable weights. The residual shortcut exists within the U-Net structure, connecting the input layer and the output layer with the same size.

To achieve conditioned image synthesis and relieve computational resources, we encode the input signal $y$ with specific encoder $\tau_\theta(\cdot)$ and train the diffusion model in latent space:

$$
\begin{aligned}
\mathcal{L}_t^{\text{LDM}} &= \mathbb{E}_{\mathcal{E}(x_0), t \sim [1,T], \epsilon_t \sim \mathcal{N}(0, \mathbf{I})}\left[\|\epsilon_t - \epsilon_\theta(\mathbf{z}_t, t, \tau_\theta(y))\|^2\right] \\
&= \mathbb{E}_{\mathcal{E}(x_0), t \sim [1,T], \epsilon_t \sim \mathcal{N}(0, \mathbf{I})}\left[\left\|\epsilon_t - \epsilon_\theta\left(\sqrt{\bar{\alpha}_t}\mathbf{z}_0 + \sqrt{1 - \bar{\alpha}_t}\epsilon_t, t, \tau_\theta(y)\right)\right\|^2\right],
\end{aligned}
\tag{13}
$$

where $\mathcal{E}(\cdot)$ is the pre-trained autoencoder, which encodes the high-dimensional pixel into low-dimensional latent space.

*3.3. Text Prompt Expansion*

Human-generated text prompts often lack critical descriptions necessary for effective image generation. If the prompt description is deficient or the correlation between keywords is less, the generated image will be particularly poor, which is unsatisfying and far from expectations. To overcome this challenge, we utilize the pre-trained large language model, GPT-2 [9], to complete and enrich the input prompt. The GPT-2 model is finetuned on the 80k text prompts collected from [27] for supporting the text-to-image task. Given any text prompt, the pre-trained large language model can complete text prompts and generate complementary content. We only need to give one concise sentence or several keywords, and the pre-trained large language model continues writing based on the prompt input rapidly. The complementary prompt improves the quality of the generated image greatly.

The flowchart of our system is shown in Figure 6. In the training stage, the latent diffusion model, which takes text as input and outputs the image, is trained with pre-trained weights [28] on a specific artist's paintings to obtain specialized weights. The text prompt expansion module, which takes text as input and outputs expanded text, is also

trained with pre-trained weights [29] to obtain general weights. Both models are trained independently. After training, we can generate a high-quality painting with a simple sentence in the testing stage.

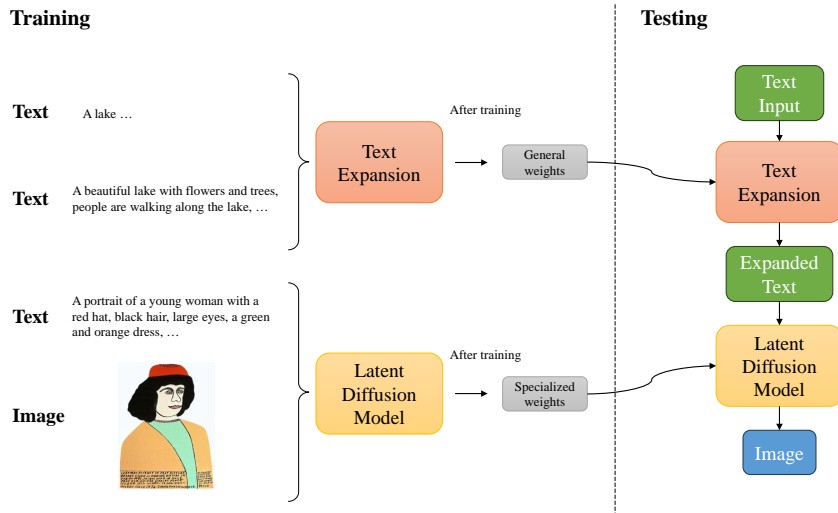

**Figure 6.** The flowchart of our system.

## 4. Experiments

### 4.1. Dataset

The *WikiArt* painting dataset collects 81,444 paintings from 1119 artists around the world, consisting of multiple different genres, styles, and art fields. The *WikiArt* dataset is the largest painting dataset that is available for research. Most paintings span from the 13th century to the 21st century. Each digital image is labeled with information regarding the artist, genre, style, and year. However, one notable shortcoming of the *WikiArt* dataset is the absence of contextual descriptions for the paintings. To generate the lacking text description, we employ the pre-trained image tagging model called RAM [30] and the vision-language model called Tag2Text [31] to generate the corresponding image text description automatically. However, wrong descriptions occasionally happen. To remove the wrong image text description, we utilize a contrastive pre-trained model [32] to calculate the similarity between the actual images and generated image text descriptions. Any image–text pair with a similarity score lower than 0.2 is subsequently removed. Following the automated text generation and selection process, we take extra measures to manually review inaccurate text descriptions, ensuring the dataset's overall quality. The number of paintings contributed by each artist is shown in Table 1. Additionally, we have included examples of images and corresponding text prompts in Figure 7, to provide a visual glimpse of the dataset.

**Table 1.** The number of paintings by each artist.

| Artist | Number |
| :---: | :---: |
| Vincent van Gogh | 1893 |
| Leonardo da Vinci | 204 |
| Pablo Picasso | 764 |
| Claude Monet | 1343 |
| Theodore Rousseau | 144 |
| Jean-Francois Millet | 122 |
| Nicholas Roerich | 1820 |
| Paul Gauguin | 394 |
| Paul Cezanne | 582 |

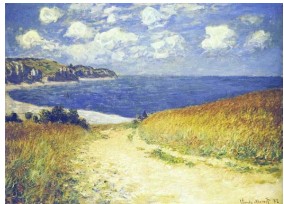

A painting of a path leading to a beach with a view of the ocean, by monet, high definition art, wikiart, hills and cliffs in the background, wheat fields, sunny day

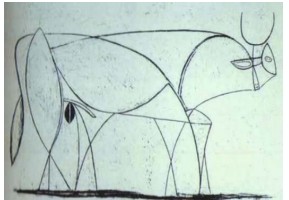

A drawing of a cow with a face and horns on a white background, by picasso, a side view of a gaunt, drawn with a single line

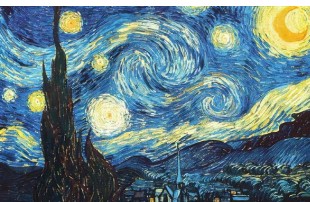

The starry night painting by vincent van gogh, twinkling and spiral nubela, castle, houses, sky, fine-art, neo-expressionism

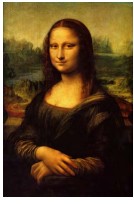

A painting of a woman with a smile, by leonardo, best painter in the world, mona lisa, greatest art ever made, the most beautiful, staring

**Figure 7.** Sample images and text descriptions of *WikiArt*.

### 4.2. Implemental Details

In our scenario, collecting training data presents a formidable challenge, compounded by the fact that the number of artworks by these late artists remains fixed. Training the latent diffusion model from scratch on the limited dataset is non-trivial, which may lead to a significant degradation in the model's performance. To make up for the insufficient training data, we utilize the core idea of transfer learning [33]. Transfer learning involves creating a high-performance model trained with data from different domains. Therefore, we leverage the pre-trained weights from the stable diffusion v1-5 model [28] and train the latent diffusion model on an NVIDIA GeForce RTX 4090 GPU. The stable diffusion v1-5 is trained on a subset of LAION-5B [34], which has seen millions of images and has robust representation capabilities. The latent diffusion model is initialized with stable diffusion v1-5 and retrained on all paintings of 9 artists from *WikiArt* Dataset. Training the latent model with pre-trained weights is beneficial to the training process. To optimize memory usage, we adopt both 16-bit and 32-bit floating-point mixed precision to train the latent diffusion model. The batch size is set to 1 and the learning rate is set to $1 \times 10^{-5}$. For each artist, the latent diffusion model undergoes independent retraining with the resolution of $512 \times 512$ for 50 epochs.

### 4.3. Qualitative Results

We retrain the latent diffusion model on nine artists' works, subsequently leveraging this model to generate novel paintings with text prompts. Notably, we propose to generate novel paintings from famous artists' works based on the diffusion model for the first time. Consequently, our comparative analysis is restricted to our retrained model and the original latent diffusion model. Figures 8–11 show generated images from the original latent diffusion model and the retrained model on *WikiArt* dataset. The first row of Figures 8–11 shows the original generated image. Based on the retrained latent diffusion model, we can request these late famous artists to create novel paintings in their styles. From the second row to the last row, the paintings of Vincent van Gogh, Leonardo da Vinci, Pablo Picasso, and other renowned artists are shown. It can be seen from Figures 8–11 that generated paintings effectively capture the distinctive and creative styles associated with each artist. Vincent van Gogh is famous for masterpieces like *The Starry Night* , *Sunflowers*, *Self-Portraits*, and so on. He had great enthusiasm for rural life and landscapes, particularly sunflowers and wheat fields. His paintings were characterized by vibrant and exaggerated colors, and his unique style is clearly discernible in the second row of Figures 8–11. In contrast, the fourth row of these figures unmistakably showcases the works of Pablo Picasso,

celebrated for pioneering highly exaggerated and distorted artistic techniques. Picasso's paintings often delved into cubism and surrealism, utilizing geometric shapes to structure his compositions. These deeply emotive and condensed artworks were immensely popular, making his unique style instantly recognizable.

Additionally, we give some modern prompts, such as *astronaut*, *drone*, *tank*, and *comptuer keyboard*, and from the figure we can see that the diffusion model can generate creative paintings that not only align with modern prompts, but also follow artists' distinctive styles. However, there are also some failure cases, such as generated images of text prompt *a woman holding a knife in her hand*. It appears that the diffusion model lacks sufficient training images or the text prompt is not detailed enough, causing the knife to be lost in some pictures. Simple text prompts sometimes cause unreasonable and unexpected results due to insufficient contextual description. To mitigate this issue, we propose to use the pre-trained language model for automatic text prompt expansion. Table 2 provides a detailed presentation of the expanded text prompts, generated with different random seeds. The visual results, shown in the second and fourth columns of Figures 8–11, clearly illustrate the significant improvements achieved through text expansion. These expanded prompts infuse the generated images with intricate detail and an overall boost in quality.

**Table 2.** Expanded text prompts.

| Text Input | An Astronaut Riding a Horse |
| --- | --- |
| Original | An astronaut in a white suit riding a horse, colorful sky in the background, epic lighting |
| Vincent van Gogh | An astronaut riding a blue horse, the upper part of a spacesuit is white and the lower part is yellow, vibrant high-contrast coloring |
| Leonardo da Vinci | An astronaut riding a horse in space, an astronaut in a triumphant pose, a closeup shot, amazing artwork, comic cover art, high spirits |
| Pablo Picasso | An astronaut riding a horse with a helmet on, horse is dyed in many colors |
| Claude Monet | A painting of a man riding a horse with a space suit on, the horse in a running state |
| Theodore Rousseau | An astronaut riding a horse on the moon, the sky full of stars is behind the astronaut, hard shadows and strong rim light |
| Jean-Francois Millet | An astronaut riding a horse, the horse is charging, the sky is filled with clouds, it seems that a storm is coming, masterpiece |
| Nicholas Roerich | An astronaut riding a horse, planets in the background, an astronaut in colorful clothes, bright light, lunar walk, vibrant colors |
| Paul Gauguin | An astronaut riding a horse over water, an astronaut with a backpack on his back, floating bubbles |
| Paul Cezanne | An astronaut riding a horse in a blue field with orange sky, rainy day, the horse is walking slowly |

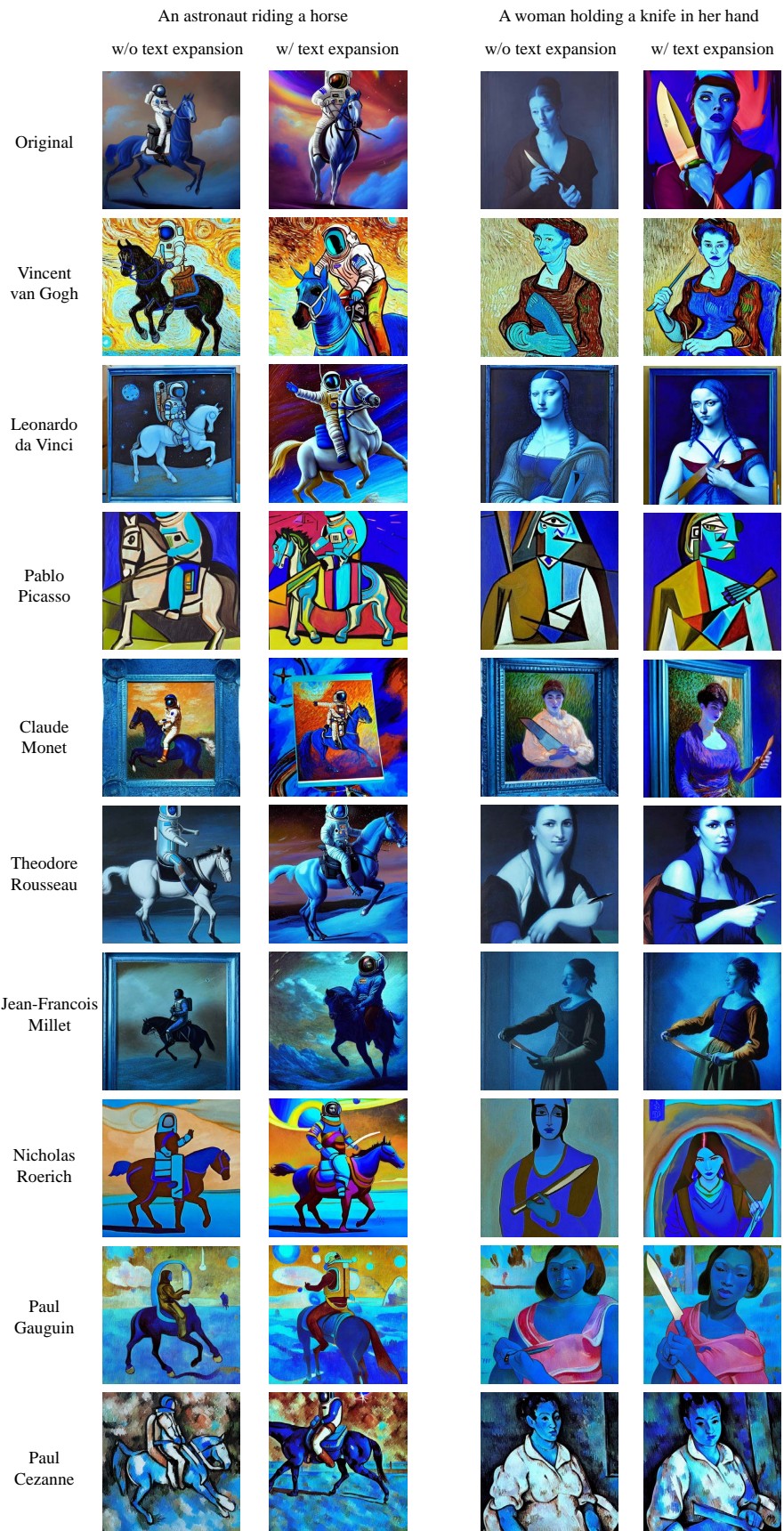

**Figure 8.** The generated novel paintings of input text prompts "An astronaut riding a horse" and "A woman holding a knife in her hand".

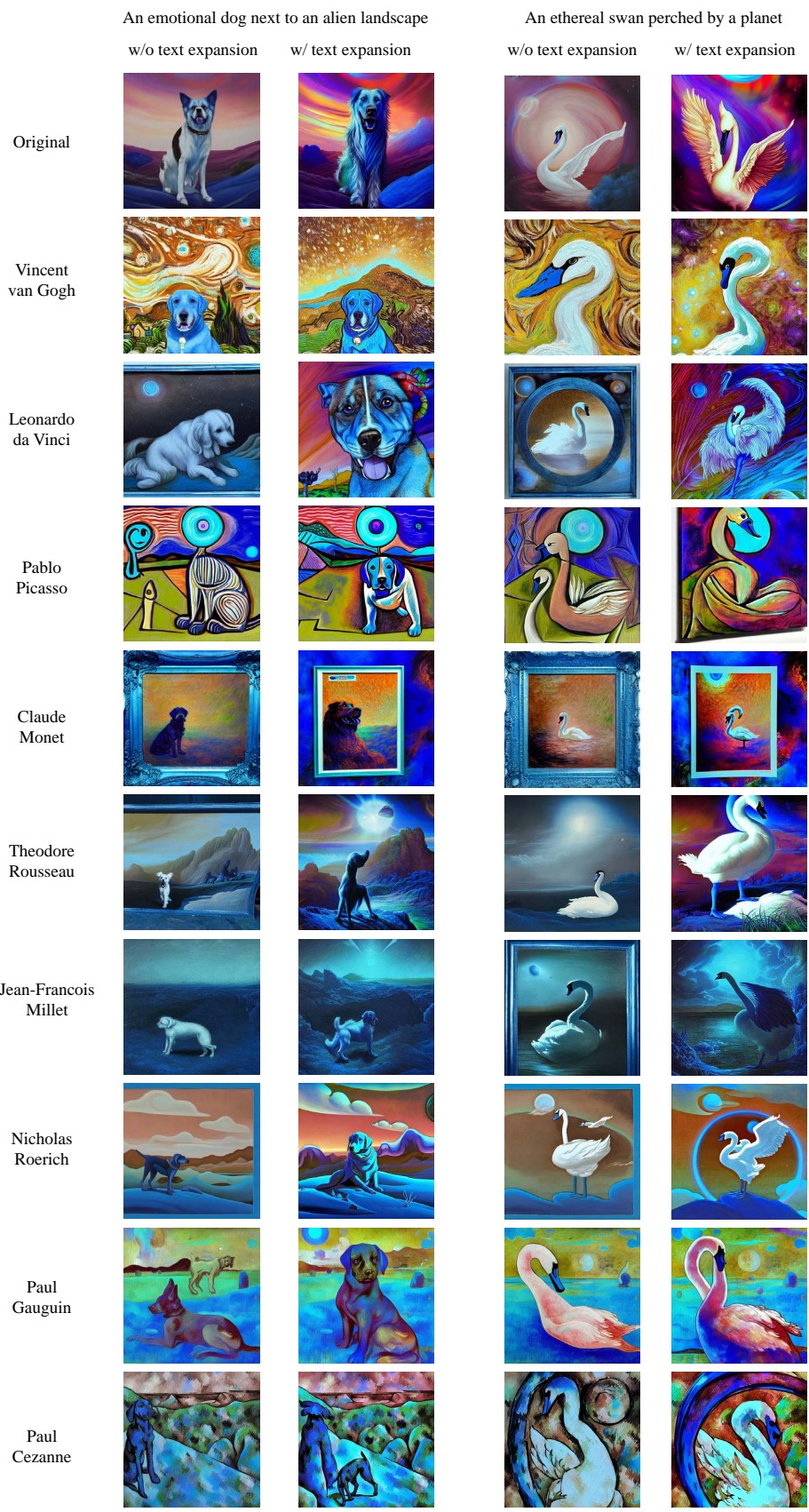

**Figure 9.** The generated novel paintings of input text prompts "An emotional dog next to an alien landscape" and "An ethereal swan perched by a planet".

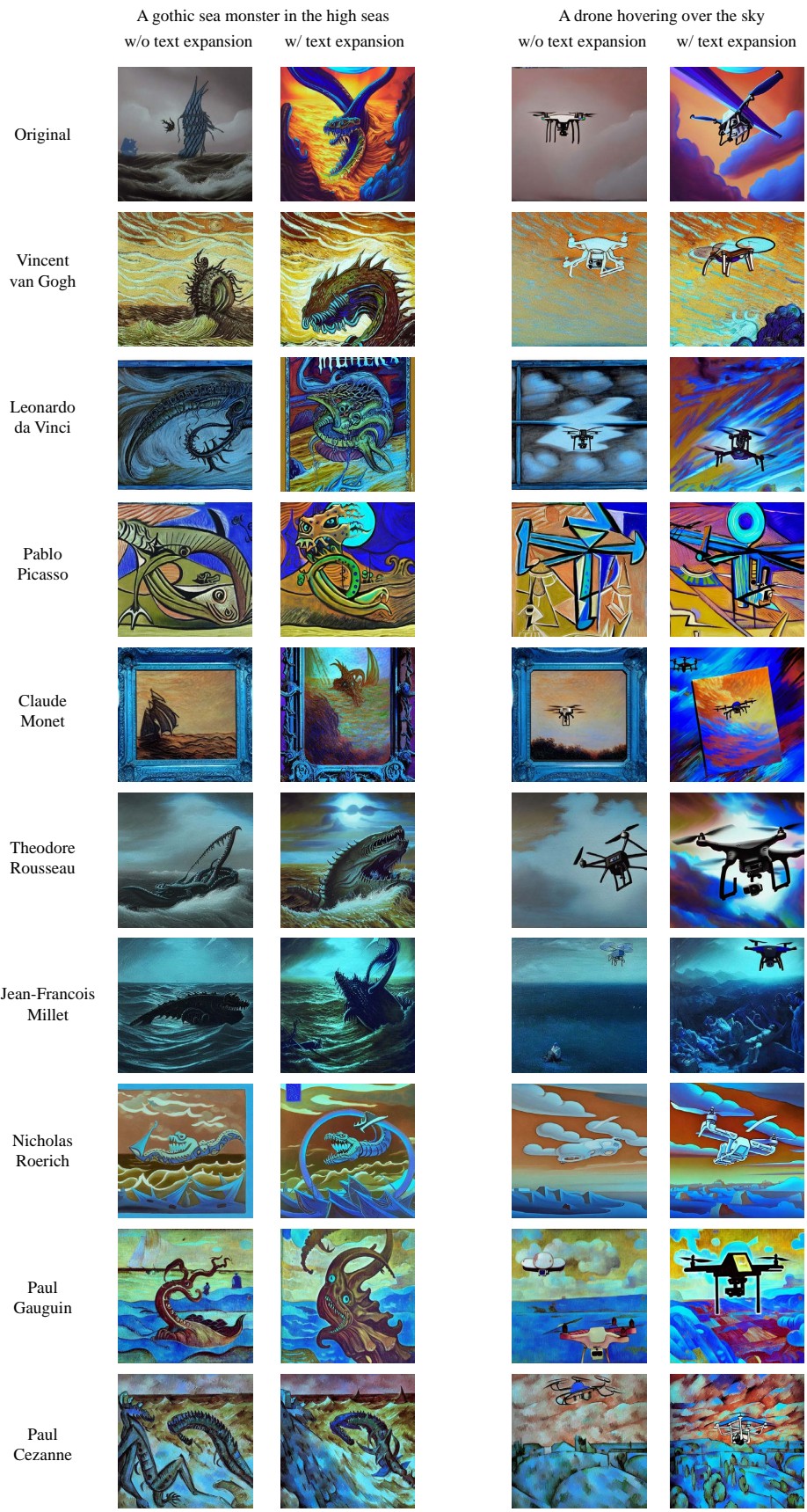

**Figure 10.** The generated novel paintings of input text prompts "A gothic sea monster in the high seas" and "A drone hovering over the sky".

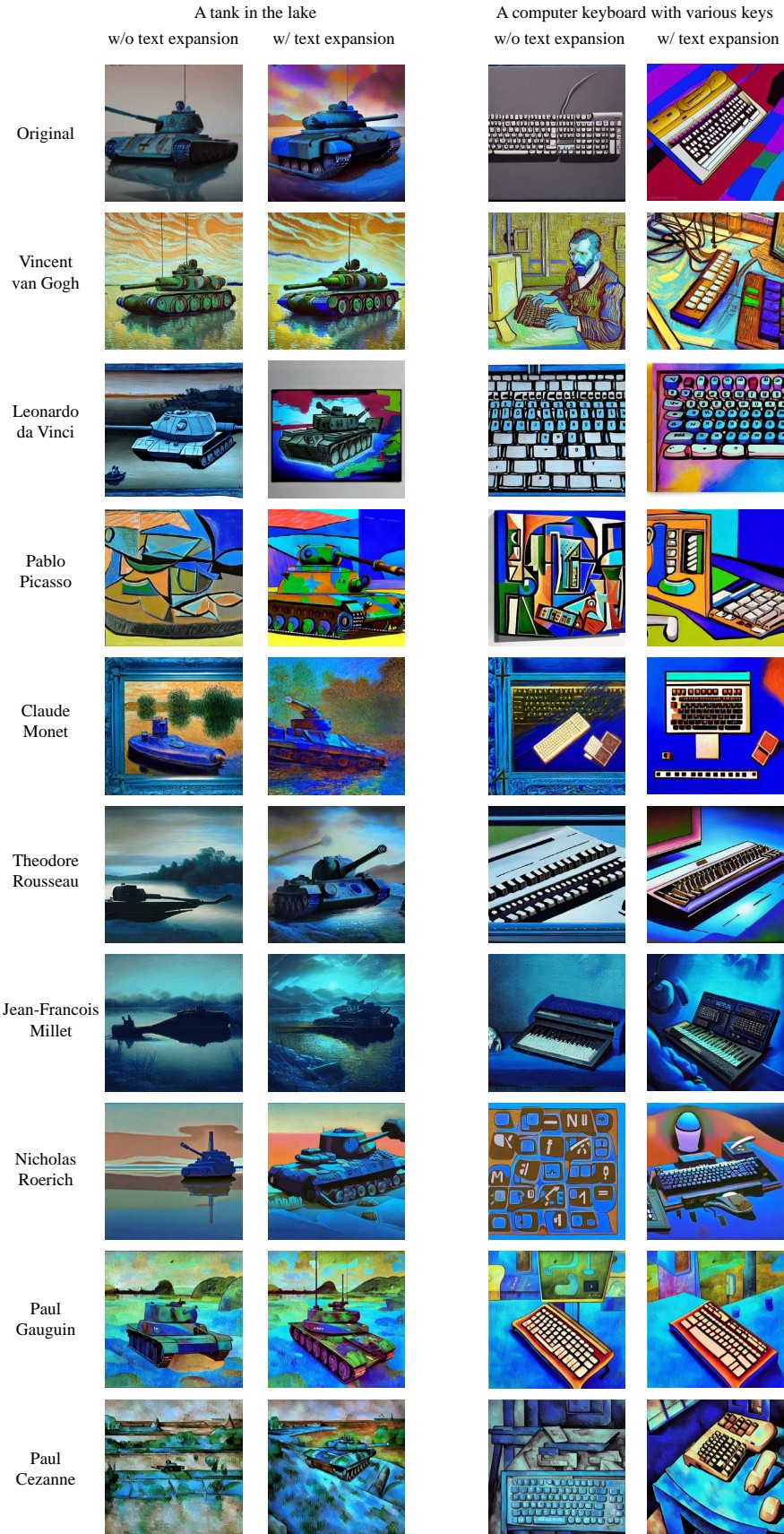

**Figure 11.** The generated novel paintings of input text prompts "A tank in the lake" and "A computer keyboard with various keys".

*4.4. Quantitative Results*

Though pairs of text prompts and images are available, the latent diffusion model may produce different results depending on different latent codes. Furthermore, the provided text prompt lacks corresponding image ground truth. Therefore, it is unreasonable to evaluate the model with pixel-aligned metrics, such as PSNR and LPIPS. We employ Fréchet Inception Distance (FID) to measure the distribution distance between generated paintings and real paintings. Quantitative results are shown in Table 3. Obviously, the retrained model achieves more realistic results compared with the original model. With the utilization of text prompt expansion, both the original model and retrained model demonstrate better performance.

**Table 3.** Quantitative comparison on FID.

|  | FID $\downarrow$ |
|---|---|
| Original w/o text expansion | 397.0344 |
| Original w/ text expansion | 366.8111 |
| Ours w/o text expansion | 290.1978 |
| Ours w/ text expansion | **284.5017** [1] |

[1] Bold indicates that the model gives the best results.

## 5. Discussion and Conclusions

In this paper, we propose to generate novel paintings from existing paintings of famous artists and demonstrate high-quality images based on the powerful latent diffusion model. Our method is dedicated to creating novel paintings by retraining the latent diffusion model and adopting text prompt expansion. After retraining on hundreds of paintings, our framework can generate artworks with modern themes. Experimental results show the effectiveness of our retrained model and text prompt expansion. However, there are still some problems with generated images. Unreasonable and unexpected results sometimes happen, which may be caused by incorrect image text descriptions in the training dataset, insufficient text prompts in the inference stage, or other reasons. Therefore, we will continue to research and develop the framework to work out these problems in the future.

The proposed framework will promote the development of image synthesis and painting creation, which has many applications, such as entertainment and movies. However, efficient painting synthesis may raise issues such as copyright and portrait rights, which may cause some legal disputes and ethical concerns. We suggest that policymakers should establish a strict regulatory system to supervise the practical use of this technology.

**Author Contributions:** S.S. proposed the initial idea; D.W. and C.M. conducted the experiment; D.W. and S.S. improved the model further; D.W. and C.M. wrote the manuscript; S.S. revised the manuscript. All authors have read and agreed to the published version of the manuscript.

**Funding:** This research was funded by the Natural Science Foundation of Heilongjiang Province, China (Grant No. LH2023E092).

**Informed Consent Statement:** Not applicable.

**Data Availability Statement:** The image and text context description dataset used in our paper is available at https://drive.google.com/file/d/1TKVkctNoYdDQPeksAndkrOk6FE7s-qcp/view?usp=sharing, accessed on 5 March 2023.

**Conflicts of Interest:** The authors declare no conflict of interest.

## Abbreviations

| | |
|---|---|
| DOAJ | Directory of open access journals |
| TLA | Three letter acronym |
| LD | Linear dichroism |

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
