# Peer review of "Novel Paintings from the Latent Diffusion Model through Transfer Learning"

_applsci, doi:10.3390/app131810379_

Round 1

Reviewer 1 Report

The article is interesting and fairly well-prepared. However, there are some revisions that should be made before publication:

  • The authors have not provided any comparison to state-of-the-art methods. FID (as a numerical metric) provides an opportunity to evaluate the proposed solution and compare it with previously proposed methods. Without such a comparison, we cannot determine whether the presented system is effective or not.

  • The addition of a flowchart presenting the system's pipeline could add value.

  • The generation of images raises significant ethical and legal concerns. These concerns should at least be mentioned and discussed in the article.

Author Response

Thanks for your insightful and constructive comments.

Comments 1:

The authors have not provided any comparison to state-of-the-art methods. FID (as a numerical metric) provides an opportunity to evaluate the proposed solution and compare it with previously proposed methods. Without such a comparison, we cannot determine whether the presented system is effective or not.

Response 1:

To the best of our knowledge, we propose to create novel paintings from famous artists' works based on the diffusion model for the first time, and we will clarify it in the new version. Therefore, we only compare our modified model with the original latent diffusion model. Experimental results also demonstrate the effectiveness of our model.

Comments 2:

The addition of a flowchart presenting the system's pipeline could add value.

Response 2:

We will add the flowchart, which is shown in Figure 6 of the new version.

Comments 3:

The generation of images raises significant ethical and legal concerns. These concerns should at least be mentioned and discussed in the article.

Response 3:

We will add discussions about ethical and legal concerns in Chapter 5.

Thank you again for your comment, it is very helpful to us.

Reviewer 2 Report

The authors propose a new transfer learning method of the latent diffusion model (LDM) for image synthesis. LDM allows for the effective execution of various generative tasks with images and videos while significantly reducing computational requirements compared to traditional diffusion models. The general LDM is adapted for image synthesis with the WikiArt dataset. Furthermore, automated prompt engineering is used to improve the results.

The portrayal of the related work is fair. Maybe some diagrams would be useful. Line 96 introduces the noise variable "z" without any related context.

In Chapter 3, it is unclear what is the original part of LDM and what is the author's contribution, if any.

In Chapter 4 (the experiment), the authors state that some details will be "released later". At least some examples should be given. It would also be beneficial if the expanded prompts were given for the example images. Without that, it is not easy to deduce why images differ in specific ways.

It was unsurprising that the prompt "A dapper woman…" has problems. For me, the word "dapper" describes a very specific piece of clothing for a man (imagine Fred Astaire in old movies). I am probably wrong, but the ChatGPT "thinks" the same. By observing text expansion, it would probably be clear that there are conflicting terms.

The paper is difficult to read. There are a lot of unusual sentence structures, and some sentences make no sense. Some of the text in the experiment description is in the future tense.

Author Response

Thanks for your insightful and constructive comments.

Comments 1:

The portrayal of the related work is fair. Maybe some diagrams would be useful. Line 96 introduces the noise variable "z" without any related context.

Response 1:

We will add diagrams in the related work and delete the noise variable "z" of Line 96.

Comments 2:

In Chapter 3, it is unclear what is the original part of LDM and what is the author's contribution, if any.

Response 2:

In Chapter 3, we give a comprehensive description about the diffusion model and we do not add any changes to the original latent diffusion model. If we change the structure of the diffusion model, training the diffusion model from scratch needs hundreds of GPUs which is unaffordable for us. Therefore, we train the diffusion model on our collected dataset with pre-trained weights. 

Comments 3:

In Chapter 4 (the experiment), the authors state that some details will be "released later". At least some examples should be given. It would also be beneficial if the expanded prompts were given for the example images. Without that, it is not easy to deduce why images differ in specific ways.

Response 3:

In Chapter 4, we will add some examples of our used dataset, which is shown in Figure 7. The expanded prompts are given in Table 2 of the new version.

Comments 4:

It was unsurprising that the prompt "A dapper woman…" has problems. For me, the word "dapper" describes a very specific piece of clothing for a man (imagine Fred Astaire in old movies). I am probably wrong, but the ChatGPT "thinks" the same. By observing text expansion, it would probably be clear that there are conflicting terms.

Response 4:

The example prompt "A dapper woman with a knife in the hand" is indeed unreasonable, and we replace it with "A woman holding a knife in her hand".

Comments 5:

The paper is difficult to read. There are a lot of unusual sentence structures, and some sentences make no sense. Some of the text in the experiment description is in the future tense.

Response 5:

For extensive English editing comments, I have invited my native English friend to correct mistakes, especially in the introduction, method, and experiment chapter.

Thank you again for the comment, it is very helpful to us.

Round 2

Reviewer 1 Report

Flowcharts need to be described in the text. Figures 1,2,3 are not mentioned.

Author Response

Thanks for your kind comments.

Comments 1:
Flowcharts need to be described in the text. Figures 1,2,3 are not mentioned.
Response 1:
We will describe flowcharts in Section 3.3 and mention Figures 1,2,3 in Chapter 2.

Reviewer 2 Report

The authors have correctly considered my comments on the first version of the paper. The paper is suitable for publication.

Author Response

Thank you for your approval of our work.